# Model Compression with Adversarial Robustness: A Unified Optimization Framework

**Shupeng Gui**$^{\diamond,*}$, **Haotao Wang**$^{\dagger,*}$, **Haichuan Yang**$^{\diamond}$, **Chen Yu**$^{\diamond}$,
**Zhangyang Wang**$^{\dagger}$ and **Ji Liu**$^{\ddagger}$

$^{\diamond}$Department of Computer Science, University of Rochester
$^{\dagger}$Department of Computer Science and Engineering, Texas A&M University
$^{\ddagger}$Ytech Seattle AI lab, FeDA lab, AI platform, Kwai Inc
$^{\dagger}${htwang, atlaswang}@tamu.edu
$^{\diamond}${sgui2, hyang36, cyu28}@ur.rochester.edu
$^{\ddagger}$ji.liu.uwisc@gmail.com

## Abstract

Deep model compression has been extensively studied, and state-of-the-art methods can now achieve high compression ratios with minimal accuracy loss. This paper studies model compression through a **different lens**: could we compress models without hurting their robustness to adversarial attacks, in addition to maintaining accuracy? Previous literature suggested that the goals of robustness and compactness might sometimes contradict. We propose a novel *Adversarially Trained Model Compression* (**ATMC**) framework. ATMC constructs a unified constrained optimization formulation, where existing compression means (pruning, factorization, quantization) are all integrated into the constraints. An efficient algorithm is then developed. An extensive group of experiments are presented, demonstrating that ATMC obtains remarkably more favorable trade-off among model size, accuracy and robustness, over currently available alternatives in various settings. The codes are publicly available at: `https://github.com/shupenggui/ATMC`.

## 1 Introduction

**Background: CNN Model Compression** As more Internet-of-Things (IoT) devices come online, they are equipped with the ability to ingest and analyze information from their ambient environments via sensor inputs. Over the past few years, convolutional neural networks (CNNs) have led to rapid advances in the predictive performance in a large variety of tasks [1]. It is appealing to deploy CNNs onto IoT devices to interpret big data and intelligently react to both user and environmental events. However, the model size, together with inference latency and energy cost, have become critical hurdles [2–4]. The enormous complexity of CNNs remains a major inhibitor for their more extensive applications in resource-constrained IoT systems. Therefore, model compression [5] is becoming increasingly demanded and studied [6–9]. We next briefly review three mainstream compression methods: *pruning, factorization*, and *quantization*.

Pruning refers to sparsifying the CNN by zeroing out non-significant weights, e.g., by thresholding the weights magnitudes [10]. Various forms of sparsity regularization were explicitly incorporated in the training process [6, 7], including structured sparsity, e.g., through channel pruning [8, 11, 12].

Most CNN layers consist of large tensors to store their parameters [13–15], in which large redundancy exists due to the highly-structured filters or columns [13–15]. Matrix factorization was thus adopted

---

$^{*}$The first two authors Gui and Wang contributed equally and are listed alphabetically.

to (approximately) decompose large weight matrices into several much smaller matrix factors [16, 13, 17]. Combining low-rank factorization and sparse pruning showed further effectiveness [18].

Quantization saves model size and computation by reducing float-number elements to lower numerical precision, e.g., from 32 bits to 8 bits or less [19, 20]. The model could even consist of only binary weights in the extreme case [21, 22]. Beyond scalar quantization, vector quantization was widely adopted too in model compression for parameter sharing [23, 24]. [25, 26] also integrated pruning and quantization in one ADMM optimization framework.

## 1.1 Adversarial Robustness: Connecting to Model Compression?

On a separate note, the prevailing deployment of CNNs also calls for attention to their **robustness**. Despite their impressive predictive powers, the state-of-the-art CNNs remain to commonly suffer from fragility to adversarial attacks, i.e., a well-trained CNN-based image classifier could be easily fooled to make unreasonably wrong predictions, by perturbing the input image with a small, often unnoticeable variation [27–34]. Other tasks, such as image segmentation [35] and graph classification [36], were all shown to be vulnerable to adversarial attacks. Apparently, such findings put CNN models in jeopardy for security- and trust-sensitive IoT applications, such as mobile bio-metric verification.

There are a magnitude of adversarial defense methods proposed, ranging from hiding gradients [37], to adding stochasticity [38], to label smoothening/defensive distillation [39, 40], to feature squeezing [41], among many more [42–44]. A handful of recent works pointed out that those empirical defenses could still be easily compromised [27], and a few certified defenses were introduced [32, 45].

To our best knowledge, there have been few existing studies on *examining the robustness of compressed models*: most CNN compression methods are evaluated only in terms of accuracy on the (clean) test set. Despite their satisfactory accuracies, it becomes curious to us: did they sacrifice robustness as a "hidden price" paid? We ask the question: could we possibly have a compression algorithm, that can lead to compressed models that are not only accurate, but also robust?

The answer yet seems to highly non-straightforward and contextually varying, at least w.r.t. different means of compression. For example, [46] showed that sparse algorithms are not stable: if an algorithm promotes sparsity, then its sensitivity to small perturbations of the input data remains bounded away from zero (i.e., no uniform stability properties). But for other forms of compression, e.g., quantization, it seems to reduce the Minimum Description Length [47] and might potentially make the algorithm more robust. In deep learning literature, [48] argued that the tradeoff between robustness and accuracy may be inevitable for the classification task. This was questioned by [49], whose theoretical examples implied that a both accurate and robust classifier might exist, given that classifier has sufficiently large model capacity (perhaps much larger than standard classifiers). Consequently, different compression algorithms might lead to different trade-offs between robustness and accuracy. [50] empirically discovered that an appropriately higher CNN model sparsity led to better robustness, whereas over-sparsification (e.g., less than 5% non-zero parameters) could in turn cause more fragility. Although sparsification (i.e., pruning) is only one specific case of compression, the observation supports a non-monotonic relationship between mode size and robustness.

A few parallel efforts [38, 51] discussed *activation* pruning or quantization as defense ways. While potentially leading to the speedup of model inference, they have no direct effect on reducing model size and therefore are not directly "apple-to-apple" comparable to us. We also notice one concurrent work [52] combining adversarial training and weight pruning. Sharing a similar purpose, our method appears to solve a more general problem, by jointly optimizing three means of pruning, factorization quantization wr.t. adversarial robustness. Another recent work [53] studied the transferability of adversarial examples between compressed models and their non-compressed baseline counterparts.

## 1.2 Our Contribution

As far as we know, this paper describes one of the first algorithmic frameworks that connects model compression with the robustness goal. We propose a unified constrained optimization form for compressing large-scale CNNs into both compact and adversarially robust models. The framework, dubbed *adversarially trained model compression* (**ATMC**), features a seamless integration of adversarial training (formulated as the optimization objective), as well as a novel structured compression constraint that jointly integrates three compression mainstreams: pruning, factorization and quantization. An efficient algorithm is derived to solve this challenging constrained problem.

While we focus our discussion on reducing model size only in this paper, we note that ATMC could be easily extended to inference speedup or energy efficiency, with drop-in replacements of the constraint (e.g., based on FLOPs) in the optimization framework.

We then conduct an extensive set of experiments, comparing ATMC with various baselines and off-the-shelf solutions. ATMC consistently shows significant advantages in achieving competitive robustness-model size trade-offs. As an interesting observation, the models compressed by ATMC can achieve very high compression ratios, while still maintaining appealing robustness, manifesting the value of optimizing the model compactness through the robustness goal.

## 2 Adversarially Trained Model Compression

In this section, we define and solve the ATMC problem. ATMC is formulated as a constrained min-max optimization problem: the adversarial training makes the min-max objective (Section 2.1), while the model compression by enforcing certain weight structures constitutes the constraint (Section 2.2). We then derive the optimization algorithm to solve the ATMC formulation (Section 2.3).

### 2.1 Formulating the ATMC Objective: Adversarial Robustness

We consider a common white-box attack setting [30]. The white box attack allows an adversary to eavesdrop the optimization and gradients of the learning model. Each time, when an "clean" image $x$ comes to a target model, the attacker is allowed to "perturb" the image into $x'$ with an adversarial perturbation with bounded magnitudes. Specifically, let $\Delta \geq 0$ denote the predefined bound for the attack magnitude, $x'$ must be from the following set:

$$B_\infty^\Delta(x) := \{x' : \|x' - x\|_\infty \leq \Delta\}.$$

The objective for the attacker is to perturb $x'$ within $B_\infty^\Delta(x)$, such as the target model performance is maximally deteriorated. Formally, let $f(\boldsymbol{\theta}; x, y)$ be the loss function that the target model aims to minimize, where $\boldsymbol{\theta}$ denotes the model parameters and $(x, y)$ the training pairs. The adversarial loss, i.e., the training objective for the attacker, is defined by

$$f^{\mathrm{adv}}(\boldsymbol{\theta}; x, y) = \max_{x' \in B_\infty^\Delta(x)} f(\boldsymbol{\theta}; x', y) \tag{1}$$

It could be understood that the maximum (worst) target model loss attainable at any point within $B_\infty^\Delta(x)$. Next, since the target model needs to defend against the attacker, it requires to suppress the worst risk. Therefore, the overall objective for the target model to gain adversarial robustness could be expressed as $\mathcal{Z}$ denotes the training data set:

$$\min_{\boldsymbol{\theta}} \sum_{(x,y) \in \mathcal{Z}} f^{\mathrm{adv}}(\boldsymbol{\theta}; x, y). \tag{2}$$

### 2.2 Integrating Pruning, Factorization and Quantization for the ATMC Constraint

As we reviewed previously, typical CNN model compression strategies include pruning (element-level [10], or channel-level [8]), low-rank factorization [16, 13, 17], and quantization [23, 19, 20]. In this work, we aim to integrate all three into a unified, flexible structural constraint.

Without loss of generality, we denote the major operation of a CNN layer (either convolutional or fully-connected) as $x_{\mathrm{out}} = \boldsymbol{W} x_{\mathrm{in}}, \boldsymbol{W} \in \mathbb{R}^{m \times n}, m \geq n$; computing the non-linearity (neuron) parts takes minor resources compared to the large-scale matrix-vector multiplication. The basic pruning [10] encourages the elements of $\boldsymbol{W}$ to be zero. On the other hand, the factorization-based methods decomposes $\boldsymbol{W} = \boldsymbol{W}_1 \boldsymbol{W}_2$. Looking at the two options, we propose to enforce the following structure to $\boldsymbol{W}$ ($k$ is a hyperparameter):

$$\boldsymbol{W} = \boldsymbol{U}\boldsymbol{V} + \boldsymbol{C}, \quad \|\boldsymbol{U}\|_0 + \|\boldsymbol{V}\|_0 + \|\boldsymbol{C}\|_0 \leq k, \tag{3}$$

where $\|\cdot\|_0$ denotes the number of nonzeros of the augment matrix. The above enforces a novel, compound (including both multiplicative and additive) sparsity structure on $\boldsymbol{W}$, compared to existing sparsity structures directly on the elements of $\boldsymbol{W}$. Decomposing a matrix into sparse factors was studied before [54], but not in a model compression context. We further allow for a sparse

error $C$ for more flexibility, as inspired from robust optimization [55]. By default, we choose $U \in \mathbb{R}^{m \times m}, V \in \mathbb{R}^{m \times n}$ in (3).

Many extensions are clearly available for equation 3. For example, the channel pruning [8] enforces rows of $W$ to be zero, which could be considered as a specially structured case of basic element-level pruning. It could be achieved by a drop-in replacement of group-sparsity norms. We choose $\ell_0$ norm here both for simplicity, and due to our goal here being focused on reducing model size only. We recognize that using group sparsity norms in (3) might potentially be a preferred option if ATMC will be adapted for model acceleration.

Quantization is another powerful strategy for model compression [20, 21, 56]. To maximize the representation capability after quantization, we choose to use the *nonuniform* quantization strategy to represent the nonzero elements in DNN parameter, that is, each nonzero element of the DNN parameter can only be chosen from a set of a few values and these values are not necessarily evenly distributed and need to be optimized. We use the notation $| \cdot |_0$ to denote the number of different values except $0$ in the augment matrix, that is,

$$|M|_0 := |\{M_{i,j} : M_{i,j} \neq 0 \ \forall i \ \forall j\}|$$

For example, for $M = [0, 1; 4; 1]$, $\|M\|_0 = 3$ and $|M|_0 = 2$. To answer the all nonzero elements of $\{U^{(l)}, V^{(l)}, C^{(l)}\}$, we introduce the non-uniform quantization strategy (i.e., the quantization intervals or thresholds are not evenly distributed). We also do not pre-choose those thresholds, but instead learn them directly with ATMC, by only constraining the number of unique nonzero values through predefining the number of representation bits $b$ in each matrix, such as

$$|U^{(l)}|_0 \leq 2^b, \ |V^{(l)}|_0 \leq 2^b, \ |C^{(l)}|_0 \leq 2^b \quad \forall l \in [L].$$

## 2.3 ATMC: Formulation

Let us use $\boldsymbol{\theta}$ to denote the (re-parameterized) weights in all $L$ layers:

$$\boldsymbol{\theta} := \{U^{(l)}, V^{(l)}, C^{(l)}\}_{l=1}^{L}.$$

We are now ready to present the overall constrained optimization formulation of the proposed ATMC framework, combining all compression strategies or constraints:

$$\min_{\boldsymbol{\theta}} \quad \sum_{(x,y) \in \mathcal{Z}} f^{\text{adv}}(\boldsymbol{\theta}; x, y) \tag{4}$$

$$\text{s.t.} \quad \underbrace{\sum_{l=1}^{L} \|U^{(l)}\|_0 + \|V^{(l)}\|_0 + \|C^{(l)}\|_0 \leq k,}_{\|\boldsymbol{\theta}\|_0} \ \text{(sparsity constraint)}$$

$$\boldsymbol{\theta} \in \mathcal{Q}_b := \{\boldsymbol{\theta} : |U^{(l)}|_0 \leq 2^b, \ |V^{(l)}|_0 \leq 2^b, \ |C^{(l)}|_0 \leq 2^b \ \forall l \in [L]\}.\text{(quantization constraint)}$$

Both $k$ and $b$ are hyper-parameters in ATMC: $k$ controls the overall sparsity of $\boldsymbol{\theta}$, and $b$ controls the quantization bit precision per nonzero element. They are both "global" for the entire model rather than layer-wise, i.e., setting only the two hyper-parameters will determine the final compression. We note that it is possible to achieve similar compression ratios using different combinations of $k$ and $b$ (but likely leading to different accuracy/robustness), and the two can indeed collaborate or trade-off with each other to achieve more effective compression.

## 2.4 Optimization

The optimization in equation 4 is a constrained optimization with two constraints. The typical method to solve the constrained optimization is using projected gradient descent or projected stochastic gradient descent, if the projection operation (onto the feasible set defined by constraints) is simple enough. Unfortunately, in our case, this projection is quite complicated, since the intersection of the sparsity constraint and the quantization constraint is complicated. However, we notice that the projection onto the feasible set defined by each individual constraint is doable (the projection onto the sparsity constraint is quite standard, how to do efficient projection onto the quantization constraint

defined set will be clear soon). Therefore, we apply the ADMM [57] optimization framework to split these two constraints by duplicating the optimization variable $\boldsymbol{\theta}$. First the original optimization formulation equation 4 can be rewritten as by introducing one more constraint

$$\min_{\|\boldsymbol{\theta}\|_0 \leq k,\ \boldsymbol{\theta}' \in \mathcal{Q}_b} \sum_{(x,y) \in \mathcal{Z}} f^{\mathrm{adv}}(\boldsymbol{\theta}; x, y) \tag{5}$$

$$\text{s.t.} \quad \boldsymbol{\theta} = \boldsymbol{\theta}'.$$

It can be further cast into a minimax problem by removing the equality constraint $\boldsymbol{\theta} = \boldsymbol{\theta}'$:

$$\min_{\|\boldsymbol{\theta}\|_0 \leq k,\ \boldsymbol{\theta}' \in \mathcal{Q}_b} \max_{\boldsymbol{u}} \sum_{(x,y) \in \mathcal{Z}} f^{\mathrm{adv}}(\boldsymbol{\theta}; x, y) + \rho \langle \boldsymbol{u}, \boldsymbol{\theta} - \boldsymbol{\theta}' \rangle + \frac{\rho}{2} \|\boldsymbol{\theta} - \boldsymbol{\theta}'\|_F^2 \tag{6}$$

where $\rho > 0$ is a predefined positive number in ADMM. Plug the form of $f^{\mathrm{adv}}$, we obtain a complete minimax optimization

$$\min_{\|\boldsymbol{\theta}\|_0 \leq k,\ \boldsymbol{\theta}' \in \mathcal{Q}_b} \max_{\substack{\boldsymbol{u}, \\ \{x^{\mathrm{adv}} \in B_\infty^\Delta(x)\}_{(x,y) \in \mathcal{Z}}}} \sum_{(x,y) \in \mathcal{Z}} f(\boldsymbol{\theta}; x', y) + \rho \langle \boldsymbol{u}, \boldsymbol{\theta} - \boldsymbol{\theta}' \rangle + \frac{\rho}{2} \|\boldsymbol{\theta} - \boldsymbol{\theta}'\|_F^2 \tag{7}$$

ADMM essentially iteratively minimizes variables $\boldsymbol{\theta}$ and $\boldsymbol{\theta}'$, and maximizes $\boldsymbol{u}$ and all $x^{\mathrm{adv}}$.

**Update $\boldsymbol{u}$**    We update the dual variable as $\boldsymbol{u}_{t+1} = \boldsymbol{u}_t + (\boldsymbol{\theta} - \boldsymbol{\theta}')$, which can be considered to be a gradient ascent step with learning rate $1/\rho$.

**Update $x^{\mathbf{adv}}$**    We update $x^{\mathrm{adv}}$ for sampled data $(x, y)$ by

$$x^{\mathrm{adv}} \leftarrow \mathrm{Proj}_{\{x':\|x'-x\|_\infty \leq \Delta\}} \left\{ x + \alpha \nabla_x f(\boldsymbol{\theta}; x, y) \right\}$$

**Update $\boldsymbol{\theta}$**    The first step is to optimize $\boldsymbol{\theta}$ in equation 7 (fixing other variables) which is only related to the sparsity constraint. Therefore, we are essentially solving

$$\min_{\boldsymbol{\theta}} \quad f(\boldsymbol{\theta}; x^{\mathrm{adv}}, y) + \frac{\rho}{2} \|\boldsymbol{\theta} - \boldsymbol{\theta}' + \boldsymbol{u}\|_F^2 \quad \text{s.t.} \ \|\boldsymbol{\theta}\|_0 \leq k.$$

Since the projection onto the sparsity constraint is simply enough, we can use the projected stochastic gradient descent method by iteratively updating $\boldsymbol{\theta}$ as

$$\boldsymbol{\theta} \leftarrow \mathrm{Proj}_{\{\boldsymbol{\theta}'':\|\boldsymbol{\theta}''\|_0 \leq k\}} \left( \boldsymbol{\theta} - \gamma \nabla_{\boldsymbol{\theta}} \left[ f(\boldsymbol{\theta}; x^{\mathrm{adv}}, y) + \frac{\rho}{2} \|\boldsymbol{\theta} - \boldsymbol{\theta}' + \boldsymbol{u}\|_F^2 \right] \right).$$

$\{\boldsymbol{\theta}'' : \|\boldsymbol{\theta}''\|_0 \leq k\}$ denotes the feasible domain of the sparsity constraint. $\gamma_t$ is the learning rate.

**Update $\boldsymbol{\theta}'$**    The second step is to optimize equation 7 with respect to $\boldsymbol{\theta}'$ (fixing other variables), which is essentially solving the following projection problem

$$\min_{\boldsymbol{\theta}'} \quad \|\boldsymbol{\theta}' - (\boldsymbol{\theta} + \boldsymbol{u})\|_F^2, \quad \text{s.t.} \ \boldsymbol{\theta}' \in \mathcal{Q}_b. \tag{8}$$

To take a close look at this formulation, we are essentially solving the following particular one dimensional clustering problem with $2^b + 1$ clusters on $\boldsymbol{\theta} + \boldsymbol{u}$ (for each $\boldsymbol{U}^{(l)}$, $\boldsymbol{V}^{(l)}$, and $\boldsymbol{C}^{(l)}$)

$$\min_{\boldsymbol{U}, \{\boldsymbol{a}_k\}_{k=1}^{2^b}} \quad \|\boldsymbol{U} - \bar{\boldsymbol{U}}\|_F^2 \quad \text{s.t.} \quad \boldsymbol{U}_{i,j} \in \{0, \boldsymbol{a}_1, \boldsymbol{a}_2, \cdots, \boldsymbol{a}_{2^b}\}.$$

The major difference from the standard clustering problem is that there is a constant cluster 0. Take $\boldsymbol{U}'^{(l)}$ as an example, the update rule of $\boldsymbol{\theta}'$ is $\boldsymbol{U}_t'^{(l)} = \mathrm{ZeroKmeans}_{2^b}(\boldsymbol{U}^{(l)} + \boldsymbol{u}_{\boldsymbol{U}^{(l)}})$, where $\boldsymbol{u}_{\boldsymbol{U}^{(l)}}$ is the dual variable with respect to $\boldsymbol{U}^{(l)}$ in $\boldsymbol{\theta}$. Here we use a modified Lloyd's algorithm [58] to solve equation 8. The detail of this algorithm is shown in Algorithm 1,

We finally summarize the full ATMC algorithm in Algorithm 2.

## 3 Experiments

To demonstrate that ATMC achieves remarkably favorable trade-offs between robustness and model compactness, we carefully design experiments on a variety of popular datasets and models as summarized in Section 3.1. Specifically, since no algorithm with exactly the same goal (adversarially robust compression) exists off-the-shelf, we craft various ablation baselines, by sequentially composing different compression strategies with the state-of-the-art adversarial training [32]. Besides, we show that the robustness of ATMC compressed models generalizes to different attackers.

Table 1: The datasets and CNN models used in the experiments.

| Models | #Parameters | bit width | Model Size (bits) | Dataset & Accuracy |
|--------|-------------|-----------|-------------------|---------------------|
| LeNet | 430K | 32 | 13,776,000 | MNIST: 99.32% |
| ResNet34 | 21M | 32 | 680,482,816 | CIFAR-10: 93.67% |
| ResNet34 | 21M | 32 | 681,957,376 | CIFAR-100: 73.16% |
| WideResNet | 11M | 32 | 350,533,120 | SVHN: 95.25% |

---

**Algorithm 1** ZeroKmeans$_B(\bar{U})$

1: **Input:** a set of real numbers $\bar{U}$, number of clusters $B$.
2: **Output:** quantized tensor $U$.
3: Initialize $a_1, a_2, \cdots, a_B$ by randomly picked nonzero elements from $\bar{U}$.
4: $Q := \{0, a_1, a_2, \cdots, a_B\}$
5: **repeat**
6:    **for** $i = 0$ to $|\bar{U}| - 1$ **do**
7:       $\delta_i \leftarrow \arg\min_j (\bar{U}_i - Q_j)^2$
8:    **end for**
9:    Fix $Q_0 = 0$
10:   **for** $j = 1$ to $B$ **do**
11:      $a_j \leftarrow \frac{\sum_i \mathbf{I}(\delta_i = j)\bar{U}_i}{\sum_i \mathbf{I}(\delta_i = j)}$
12:   **end for**
13: **until** Convergence
14: **for** $i = 0$ to $|\bar{U}| - 1$ **do**
15:   $U_i \leftarrow Q_{\delta_i}$
16: **end for**

---

**Algorithm 2** ATMC

1: **Input:** dataset $\mathcal{Z}$, stepsize sequence $\{\gamma_t > 0\}_{t=0}^{T-1}$, update steps $n$ and $T$, hyper-parameter $\rho$, $k$, and $b$, $\Delta$
2: **Output:** model $\theta$
3: $\alpha \leftarrow 1.25 \times \Delta/n$
4: Initialize $\theta$, let $\theta' = \theta$ and $u = 0$
5: **for** $t = 0$ **to** $T - 1$ **do**
6:    Sample $(x, y)$ from $\mathcal{Z}$
7:    **for** $i = 0$ to $n - 1$ **do**
8:      $x^{\text{adv}} \leftarrow \text{Proj}_{\{x':\|x'-x\|_\infty \leq \Delta\}}\{x + \alpha\nabla_x f(\theta; x, y)\}$
9:    **end for**
10:   $\theta \leftarrow \text{Proj}_{\{\theta'':\|\theta''\|_0 \leq k\}}\big(\theta - \gamma_t \nabla_\theta[f(\theta; x^{\text{adv}}, y) + \frac{\rho}{2}\|\theta - \theta' + u\|_F^2]\big)$
11:   $\theta' \leftarrow \text{ZeroKmeans}_{2^b}(\theta + u)$
12:   $u \leftarrow u + (\theta - \theta')$
13: **end for**

---

### 3.1 Experimental Setup

**Datasets and Benchmark Models** As in Table 1, we select four popular image classification datasets and pick one top-performer CNN model on each: LeNet on the MNIST dataset [59]; ResNet34 [60] on CIFAR-10 [61] and CIFAR-100 [61]; and WideResNet [62] on SVHN [63].

**Evaluation Metrics** The classification accuracies on both benign and on attacked testing sets are reported, the latter being widely used to quantify adversarial robustness, e.g., in [32]. The model size is computed via multiplying the quantization bit per element with the total number of non-zero elements, added with the storage size for the quantization thresholds ( equation 8). The compression ratio is defined by the ratio between the compressed and original model sizes. A desired model compression is then expected to achieve strong adversarial robustness (accuracy on the attacked testing set), in addition to high benign testing accuracy, at compression ratios from low to high .

**ATMC Hyper-parameters** For ATMC, there are two hyper-parameters in equation 4 to control compression ratios: $k$ in the sparsity constraint, and $b$ in the quantization constraint. In our experiments, we try 32-bit ($b = 32$) full precision, and 8-bit ($b = 8$) quantization; and then vary different $k$ values under either bit precision, to navigate different compression ratios. We recognize that a better compression-robustness trade-off is possible via fine-tuning, or perhaps to jointly search for, $k$ and $b$.

**Training Settings** For adversarial training, we apply the PGD [32] attack to find adversarial samples. Unless otherwise specified, we set the perturbation magnitude $\Delta$ to be 76 for MNIST and 4 for the other three datasets. (The color scale of each channel is between 0 and 255.) Following the settings in [32], we set PGD attack iteration numbers $n$ to be 16 for MNIST and 7 for the other three datasets. We follow [30] to set PGD attack step size $\alpha$ to be $\min(\Delta + 4, 1.25\Delta)/n$. We train ATMC for 50, 150, 150, 80 epochs on MNIST, CIFAR10, CIFAR100 and SVHN respectively.

**Adversarial Attack Settings** Without further notice, we use PGD attack with the same settings as used in adversarial training on testing sets to evaluate model robustness. In section 3.3, we also evaluate model robustness on PGD attack, FGSM attack [29] and WRM attack [45] with varying attack parameter settings to show the robustness of our method across different attack settings.

## 3.2 Comparison to Pure Compression, Pure Defense, and Their Mixtures

Since no existing work directly pursues our same goal, we start from two straightforward baselines to be compared with ATMC: standard compression (without defense), and standard defense (without compression). Furthermore, we could craft "mixture" baselines to achieve the goal: first applying a defense method on a dense model, then compressing it, and eventually fine-tuning the compressed model (with parameter number unchanged, e.g., by fixing zero elements) using the defense method again. We design the following **seven** comparison methods (the default bit precision is 32 unless otherwise specified):

- *Non-Adversarial Pruning* (**NAP**): we train a dense state-of-the-art CNN and then compress it by the pruning method proposed in [10]: only keeping the largest-magnitudes weight elements while setting others to zero, and then fine-tune the nonzero weights (with zero weights fixed) on the training set again until convergence, . NAP can thus explicitly control the compressed model size in the same way as ATMC. There is no defense in NAP.

- *Dense Adversarial Training* (**DA**): we apply adversarial training [32] to defend a dense CNN, with no compression performed.

- *Adversarial Pruning* (**AP**): we first apply the defensive method [32] to pre-train a defense CNN. We then prune the dense model into a sparse one [10], and fine-tune the non-zero weights of pruned model until convergence, similarly to NAP.

- *Adversarial $\ell_0$ Pruning* (**A$\ell_0$**): we start from the same pre-trained dense defensive CNN used by AP and then apply $\ell_0$ projected gradient descent to solve the constrained optimization problem with an adversarial training objective and a constraint on number of non-zero parameters in the CNN. Note that this is in essence a combination of one state-of-the-art compression method [64] and PGD adversarial training.

- *Adversarial Low-Rank Decomposition* (**ALR**): it is all similar to the AP routine, except that we use low rank factorization [17] in place of pruning to achieve the compression step.

- *ATMC* (**8 bits, 32 bits**): two ATMC models with different quantization bit precisions are chosen. For either one, we will vary $k$ for different compression ratios.

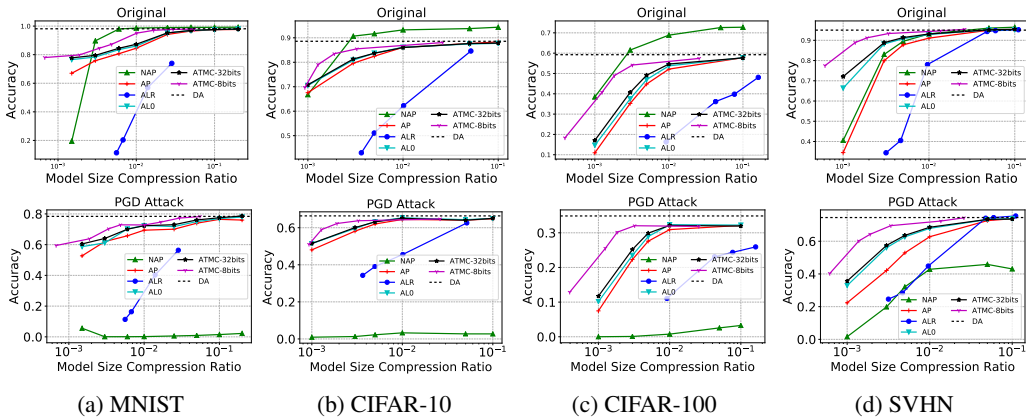

Figure 1: Comparison among NAP, AP, A$\ell_0$, ALR and ATMC (32 bits & 8 bits) on four models/datasets. **Top row**: accuracy on benign testing images versus compression ratio. **Bottom row**: robustness (accuracy on PGD attacked testing images) versus compression ratio. The black dashed lines mark the the uncompressed model results.

Fig 1 compares the accuracy on benign (top row) and PGD-attack (bottom row) testing images respectively, w.r.t. the compression ratios, from which a number of observations can be drawn.

First, our results empirically support the existence of inherent trade-off between robustness and accuracy at different compression ratios; although the practically achievable trade-off differs by method. For example, while NAP (a standard CNN compression) obtains decent accuracy results on benign testing sets (e.g., the best on CIFAR-10 and CIFAR-100), it becomes very deteriorated in terms of robustness under adversarial attacks. That verifies our motivating intuition: *naive compression, while still maintaining high standard accuracy, can significantly compromise robustness* – the "hidden price" has indeed been charged. The observation also raises a red flag for current evaluation ways of CNN compression, where the robustness of compressed models is (almost completely) overlooked.

Second, while both AP and ALR consider compression and defense in ad-hoc sequential ways, $A\ell_0$ and ATMC-32 bits further gain notably advantages over them via "joint optimization" type methods, in achieving superior trade-offs between benign test accuracy, robustness, and compression ratios. Furthermore, ATMC-32 bits outperforms $A\ell_0$ especially at the low end of compression ratios. That is owing to the the new decomposition structure that we introduced in ATMC.

Third, ATMC achieves comparable test accuracy and robustness to DA, with only minimal amounts of parameters after compression. Meanwhile, ATMC also achieves very close, sometimes better accuracy-compression ratio trade-offs on benign testing sets than NAP, with much enhanced robustness. Therefore, it has indeed combined the best of both worlds. It also comes to our attention that for ATMC-compressed models, the gaps between their accuracies on benign and attacked testing sets are smaller than those of the uncompressed original models. That seems to potentially suggest that compression (when done right) in turn has positive performance regularization effects.

Lastly, we compare between ATMC-32bits and ATMC-8bits. While ATMC-32bits already outperforms other baselines in terms of robustness-accuracy trade-off, more aggressive compression can be achieved by ATMC-8bits (with further around four-time compression at the same sparsity level), with still competitive performance. The incorporation of quantization and weight pruning/decomposition in one framework allows us to flexibly explore and optimize their different combinations.

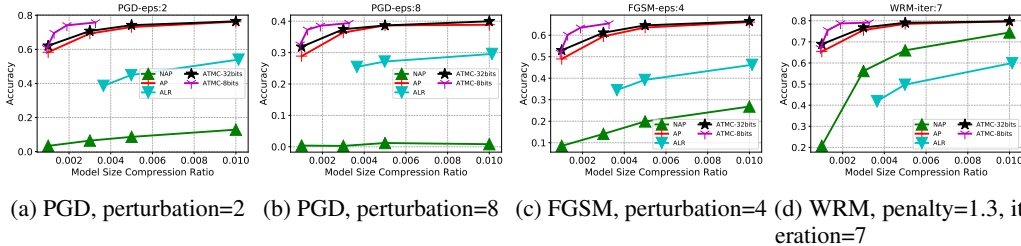

(a) PGD, perturbation=2  (b) PGD, perturbation=8  (c) FGSM, perturbation=4  (d) WRM, penalty=1.3, iteration=7

Figure 2: Robustness-model size trade-off under different attacks and perturbation levels. Note that the accuracies here are all measured by attacked images, i.e., indicating robustness.

### 3.3 Generalized Robustness Against Other Attackers

In all previous experiments, we test ATMC and other baselines against the PGD attacker at certain fixed perturbation levels. We will now show that the superiority of ATMC persists under different attackers and perturbation levels. On CIFAR-10 (whose default perturbation level is 4), we show the results against the PGD attack with perturbation levels 2 and 8, in Fig 2a and Fig 2b, respectively. We also try the FGSM attack [29] with perturbation 4, and the WRM attack [45] with penalty parameter 1.3 and iteration 7, with results displayed in Fig 2c and Fig 2d, respectively. As we can see, ATMC-32bit outperforms its strongest competitor AP in the full compression spectrum. ATMC-8bit can get more aggressively compressed model sizes while maintaining similar or better robustness to ATMC-32bit at low compression ratios. In all, the robustness gained by ATMC compression is observed to be sustainable and generalizable.

## 4 Conclusion

This paper aims to address the new problem of simultaneously achieving high robustness and compactness in CNN models. We propose the ATMC framework, by integrating the two goals in one unified constrained optimization framework. Our extensive experiments endorse the effectiveness of ATMC by observing: i) naive model compression may hurt robustness, if the latter is not explicitly taken into account; ii) a proper joint optimization could achieve both well: a properly compressed model could even maintain almost the same accuracy and robustness compared to the original one.

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
