[Supplementary Material]

# A  ATMC combined with different adversarial training methods

While we used PGD attack mainly because it is state-of-the-art, ATMC is certainly compatible with different adversarial training methods. We hereby provide results when using WRM with the same hyper-parameter settings in Section 3.3 for all training and testing. We show results with respect to the pruning ratios (PRs) (e.g, by controlling $k$ only in Equation 4). Note that for AP, A$\ell_0$ and ATMC, PRs equal standard compression ratios (CRs) if there is no quantization. Hence importantly, for ATMC-8bits, it has **only $\frac{1}{4}$ model size** compared to AP, A$\ell_0$ and ATMC-32bits, when they have the same PRs. The results are shown in Table 2.

Table 2: Results for WRM adversarial training and testing

|     | Pruning Ratio | 0.1 | 0.05 | 0.001 |
|-----|---------------|--------|--------|--------|
| TA  | AP | 0.9145 | 0.9117 | 0.7878 |
|     | A$\ell_0$ | 0.9117 | 0.9103 | 0.8206 |
|     | ATMC-32bits | **0.9156** | **0.9095** | **0.8284** |
|     | ATMC-8bits | 0.9004 | 0.9019 | 0.8109 |
| ATA | AP | 0.8271 | 0.8190 | 0.6952 |
|     | A$\ell_0$ | 0.8250 | 0.8175 | 0.7262 |
|     | ATMC-32bits | **0.8331** | **0.8289** | **0.7311** |
|     | ATMC-8bits | 0.8112 | 0.7996 | 0.7144 |

The advantage of ATMC also persists for different $\epsilon$ in PGD adversarial training. For example, if we change $\epsilon$ to 8 on CIFAR-10 dataset for both adversarial training and testing, while keeping other settings untouched, results would be like Table 3.

Table 3: Results for PGD ($\epsilon = 8$) adversarial training and testing

|     | Pruning Ratio | 0.005 | 0.003 | 0.001 |
|-----|---------------|--------|--------|--------|
| TA  | AP | 0.7296 | 0.6905 | 0.5510 |
|     | A$\ell_0$ | 0.7563 | 0.7169 | 0.5564 |
|     | ATMC-32bits | **0.7569** | **0.7219** | **0.5678** |
|     | ATMC-8bits | 0.7486 | 0.7168 | 0.5588 |
| ATA | AP | 0.4569 | 0.4247 | 0.3398 |
|     | A$\ell_0$ | 0.4813 | 0.4559 | 0.3532 |
|     | ATMC-32bits | **0.4875** | **0.4645** | **0.3608** |
|     | ATMC-8bits | 0.481 | 0.4486 | 0.3529 |

# B  ATMC quantization vs traditional quantization

ATMC jointly learn pruning and non-uniform quantization though ADMM algorithm. To further show its advantage, we compare ATMC-8bits with another baseline, that first applies ATMC-32bits and then quantizes to 8 bits using standard uniform quantization as post-processing (denoted as ATMC-8bits-uniform). The results on SVHN are shown in Table 4.

Table 4: ATMC quantization vs traditional quantization

|     | Pruning Ratio | 0.01 | 0.001 |
|-----|---------------|--------|--------|
| TA  | ATMC-8bits | **0.9336** | **0.7749** |
|     | ATMC-8bits-uniform | 0.9311 | 0.7239 |
| ATA | ATMC-8bits | **0.6954** | **0.4028** |
|     | ATMC-8bits-uniform | 0.6855 | 0.3533 |

# C Experiments for larger Neural Networks

We present results with CIFAR-10 on ResNet101 [60] in Table 5. Other experimental settings are identical with Section 3.1.

Table 5: Results on ResNet101

|  |  | 0.005 | 0.001 | 0.0008 |
|---|---|---|---|---|
| | Pruning Ratio | 0.005 | 0.001 | 0.0008 |
| TA | AP | 0.8543 | 0.6232 | 0.5599 |
| | $A\ell_0$ | 0.8621 | 0.6750 | 0.6424 |
| | ATMC-32bits | **0.8796** | **0.7345** | **0.7110** |
| ATA | AP | 0.5964 | 0.3859 | 0.3254 |
| | $A\ell_0$ | 0.6124 | 0.4263 | 0.4024 |
| | ATMC-32bits | **0.6219** | **0.4477** | **0.4347** |