[Reviews · NeurIPS 2019]

Reviewer 1



Some experimental details will need to be clarified in rebuttal: - "Unless otherwise specified, we set the perturbation magnitude to be 76 for MNIST and 4 for the other three datasets" why choose those specific magnitudes? - "We set PGD attack iteration numbers n to be 16 for MNIST and 7 for the other three datasets" could ATML stand robust to more iterations, e.g., >= 20? - Have you used random starting to alleviate gradient masking? - Would the authors consider releasing their codes for reproducibility? - Two missing references, that both empirically studied the preservation of robustness under quantization: "Robustness of Compressed Convolutional Neural Networks", IEEE BigData 2018 "To compress or not to compress: Understanding the Interactions between Adversarial Attacks and Neural Network Compression", SysML 2019.

Reviewer 2



This paper formulates a new problem and proposed a reasonable algorithm. However, I am not totally convinced the current jointly optimization is significantly better a two steps approach of (1) first do adversarial learning and then (2) compress the model by pruning and compression. As shown in Figure 2, the performance of the proposed method is almost the same as the two step approach (adversarial learning + pruning). Moreover, the current paper could be improved in the following aspects: - eq (3): what about the non-conversational layers? - line 128-line 133: it is not clear what "the nonuniform quantization" means, and how it leads to the equation between line 132 and line 133. - eq (4): in this paper f^adv seems to be limited to PSD attack. I wish to see results of other adversarial learning. - Table 1: all the networks here are relatively small, for which the compression seems not very important. Is it possible to provide experiments for large neural networks?

Reviewer 3



The idea seems interesting. It is aligned with the recent wave of studying standard versus robust accuracy; and focuses on a specific, relatively less noticed problem field (compression). The loss of robustness was overlooked in most CNN compression literature. This paper addressed it well and would bring in new insights. The authors also spent much effort into exploring different settings of model compression and adversarial training, which is appealing in revealing the relationship between the two aspects. The writeup is mature and clear to follow; the literature review of robustness-compactness relationship (Sec 1.1) is very thorough and interesting.

[Author Response · NeurIPS 2019]

We thank all three reviewers for unanimously recognizing the novelty and merits of our work, and have addressed all
their raised concerns below. We promise to release all codes and pre-trained models upon acceptance.

## Response to R2

1. **Is jointly optimization better than two-step approaches?** The best "two-step" baseline we tested (in terms of
achieving both high accuracy and robustness) is AP (first pruning then adversarial training). Compared to AP, the
superiority of both ATMC-32 bits and ATMC-8 bits is notable and consistent across all experiments (see Fig. 1).
The other strong baseline we crafted is $A\ell_0$. It is built on a SOTA sophisticated compression scheme (ICLR'19)
(replacing hardware energy with model size as the constraint, to fit our goal). Note that $A\ell_0$ **is not a two-step method**:
we replaced the ICLR'19 original objective (accuracy-driven) with our same adversarial training objective, then
optimized from end to end: it is essentially very similar to ATMC (lines 231). Therefore, if ATMC outperforms $A\ell_0$, it
is only owing to ATMC's "novel parameterizations" of weights. We apologize if it caused any confusion for R2.
In view of above, we find ATMC-32 bits (i.e., no quantization) to constantly perform better (e.g., by 5% accuracy and 2%
robustness, for SVNH at 0.1% compression ratio) or at least comparably than $A\ell_0$. ATMC-8 bits (quantization jointly
optimized) obtains a further enlarged margin over $A\ell_0$. For another comparison, we tried to quantize $A\ell_0$-compressed
models to 8 bits, and observe notably degraded performance. On SVNH at compression ratios $\frac{1}{4}[0.01, 0.005, 0.001]$, it
leads to [0.6%, 0.4%, 11.3%] drop of accuracy, and [1.5%, 2.1%, 8.1%] drop of robustness, compared to ATMC-8 bits.

2. **What about the non-convolutional layers?** (we conjecture "non-conversational" to be typo) ATMC compresses
both convolutional and fully connected layers. The latter can be directly represented as an m-by-n matrix $W$ in Eqn. (3).

3. **Unclear about "nonuniform quantization", and equation between line 132-133.** Here we refer to element
quantization whose quantization intervals are not of the same length, in contrast to using uniform (evenly distributed)
thresholds. More importantly, we do not pre-choose those intervals for quantization, but instead learn quantized matrices
$U$, $V$ and $C$ directly within ATMC, by only constraining the number of *unique* nonzero values (denoted by the equation
between line 132-133) in each matrix. We consider such jointly learned non-uniform quantization an important merit of
ATMC. To further show its advantage, we compare ATMC-8bits with another baseline, that first applies ATMC-32bits
then quantizes to 8bits (using standard uniform quantization) as post-processing. On SVNH at compression ratios
$\frac{1}{4}[0.01, 0.005, 0.001]$, it degrades both accuracy and robustness by up to **5%**, compared to ATMC-8bits.

4. $f^{\mathbf{adv}}$ **with other adversarial learning.** While we used PGD attack mainly because it is SOTA, ATMC is certainly
compatible with other attacks. We hereby provide results when using WRM [39] for all training (the robustness is also
tested with WRM attack). We show results w.r.t. the pruning ratios (PRs) (e.g, by controlling $k$ only in Eq. (4)). Note
that for AP/$A\ell_0$/ATMC, PRs equal standard compression ratios if there is no quantization (32 bits). Hence importantly,
for ATMC-8 bits, it only has **1/4 model size** compared to ATMC-32 bits/AP/$A\ell_0$, when they have the same PR.
Within the PR range [0.1, 0.05, 0.001], we obtain the accuracy (clean): **AP** [91.45%, 91.17%, 78.78%], $A\ell_0$
[91.17%, 90.03%, 82.06%], **ATMC-32bits** [91.56%, 90.95%, 82.84%], **ATMC-8bits** [90.04%, 90.19%, 81.09%];
robustness: **AP** [82.71%, 81.90%, 69.52%], $A\ell_0$ [82.50%, 81.75%, 72.62%], **ATMC-32bits** [83.31%, 82.89%, 73.11%],
**ATMC-8bits** [81.12%, 79.96%, 71.44%]. As we observe: first under the same model size, ATMC-32bits consistently
outperforms AP/$A\ell_0$; then with only 1/4 model sizes (same PRs), ATMC-8bits yields highly competitive results to 32
bits. We also observed generalized robustness of ATMC to other attackers. We will include all results in camera-ready.

5. **Experiments for large NNs?** We present results with CIFAR-10 on ResNet101 at PRs [0.005, 0.001, 0.0008]. We
obtain accuracy (clean): **AP** [85.43%, 62.32%, 55.99%], **ATMC-32bits** [86.21%, 67.50%, 64.24%], robustness: **AP**
[59.64%, 38,59%, 32.54%], **ATMC-32bits** [61.24%, 42.63%, 40.24%], Those preliminary results endorse ATMC's
effectiveness for large CNNs. More comparisons will be reported in camera-ready.

## Response to R1 and R3

1. **Attack magnitudes, and more iterations (R1):** MNIST is relatively easy so we follow [26] to use a large
perturbation 76. For other three datasets, we show magnitude 4 as an example, while the advantage of ATMC persists
in the wide range of magnitudes we tried. For example, if we change the magnitude to 8 on CIFAR-10, then at PRs
[0.01, 0.005, 0.001], we have: accuracy (clean): **AP** [77.46%, 72.96%, 55.10%], **ATMC-32bits** [78.94%, 75.69%,
56.78%]; robustness: **AP** [48.83%, 45.69%, 33.98%], **ATMC-32bits** [50,28%, 48.75%, 36.08%]. Further, at the same
group of PRs (but with only $1/4$ above corresponding sizes), **ATMC-8bits** has accuracy [78.99%, 74.86%, 55.88%];
and robustness [48.60%, 48.10%, 35.29%].
We also confirm that ATMC stands robust beyond 20 iterations. For example, on CIFAR-10 with PRs [0.01, 0.005,
0.001] against 40-iteration PGD attacks, we have the robustness of **ATMC-32bits** [64.35%, 62.44%, 51.72%], still
outperforming other baselines in the same setting. Correspondingly at the same group of PRs (thus with 1/4 sizes),
**ATMC-8bits** has robustness [62.99%, 61.55%, 50.65%]. We will include all those results in camera-ready.

2. **Miscellaneous (R1 + R3):** 1) Yes, we used random starting in all experiments; 2) We will add missing references; 3)
Compared to NAP (simple pruning), the training time of ATMC is several times longer. Compared to other adversarial
learning baselines (AP, $A\ell_0$), it is comparable; 4) One unified controlling parameter is a great idea: we will try in future.

[Meta-Review · NeurIPS 2019]

Good paper. Accept. Please include the additional results in the final version of the paper.